# Predicting Mortality in Patients with Atrial Fibrillation and Obstructive Chronic Coronary Syndrome: The Bialystok Coronary Project

**DOI:** 10.3390/jcm10214949

**Published:** 2021-10-26

**Authors:** Łukasz Kuźma, Anna Tomaszuk-Kazberuk, Anna Kurasz, Sławomir Dobrzycki, Marek Koziński, Bożena Sobkowicz, Gregory Y. H. Lip

**Affiliations:** 1Department of Invasive Cardiology, Medical University of Białystok, 15-089 Białystok, Poland; kuzma.lukasz@gmail.com (Ł.K.); annaxkurasz@gmail.com (A.K.); slawek_dobrzycki@yahoo.com (S.D.); 2Department of Cardiology, Medical University of Białystok, 15-089 Białystok, Poland; sobkowic@wp.pl; 3Department of Cardiology and Internal Medicine, Medical University of Gdańsk, 81-519 Gdynia, Poland; marek.kozinski@gumed.edu.pl; 4Liverpool Centre for Cardiovascular Science, University of Liverpool and Liverpool Heart & Chest Hospital, Liverpool L14 3PE, UK; gregory.lip@liverpool.ac.uk; 5Aalborg Thrombosis Research Unit, Department of Clinical Medicine, Aalborg University, 9220 Aalborg, Denmark

**Keywords:** atrial fibrillation, chronic coronary syndrome, coronary artery disease, mortality, AF-CAD study

## Abstract

Over the next decades, the prevalence of atrial fibrillation (AF) is estimated to double. Our aim was to investigate the causes of the long-term mortality in relation to the diagnosis of atrial fibrillation (AF) and chronic coronary syndrome (CCS). The analysed population consisted of 7367 consecutive patients referred for elective coronary angiography enrolled in a large single-centre retrospective registry, out of whom 1484 had AF and 2881 were diagnosed with obstructive CCS. During follow-up (median = 2029 days), 1201 patients died. The highest all-cause death was seen in AF(+)/CCS(+) [194/527; 36.8%], followed by AF(+)/CCS(−) [210/957; 21.9%], AF(−)/CCS(+) [(459/2354; 19.5%)] subgroups. AF ([HR]_AC_ = 1.48, 95%CI, 1.09–2.01; HR_CV_ = 1.34, 95%CI, 1.07–1.68) and obstructive CCS (HR_AC_ = 1.90, 95%CI, 1.56–2.31; HR_CV_ = 2.27, 95%CI, 1.94–2.65) together with age, male gender, heart failure, obstructive pulmonary disease, diabetes were predictors of both all-cause and CV mortality. The main findings are as follow among patients referred for elective coronary angiography, both AF and obstructive CCS are strong and independent predictors of the long-term mortality. Mortality of AF without CCS was at least as high as non-AF patients with CCS. CV deaths were more frequent than non-CV deaths in AF patients with CCS compared to those with either AF or CCS alone.

## 1. Introduction

The prevalence of atrial fibrillation (AF) in adults varies from 2% to 4%, but about one of three cases remains undetected [1,2,3]. Over the next few years, the prevalence of AF is estimated to double, associated with the aging of the population and the increasing incidence of hypertension, diabetes, heart failure, mitral valve defects, and chronic coronary syndromes (CCS). Indeed, one in five patients with CCS have coexisting AF, leading to a worsening of the patient’s prognosis [4]. The risk of ischemic stroke and heart failure in AF patients with CCS, as well as reduced life expectancy, is greater than in the CCS population without the AF [5,6,7].

In the LIFE-Heart Study, there were no associations between AF and location of coronary stenosis among patients with single-vessel coronary artery disease (CAD) and, in comparison to patients with single-vessel CAD, the risk for AF was lower in those with double and triple CAD [8]. In our Białystok Coronary Project of patients undergoing elective coronary angiography, AF was associated with a lack of obstructive coronary lesions [9]. The reason for these findings might be multifactorial, such as AF symptoms may mimic CCS symptoms, computed tomography scan and stress tests are difficult to interpret in AF patients.

AF is associated with high morbidity and mortality, placing a significant burden on the patients themselves as well as the health care system [10,11]. The presence of AF alone independently increases the risk of death [12,13]. However, there is a paucity of data regarding a possible association between the diagnosis of AF and/or CCS and long-term mortality. When considering this potential relationship, numerous questions arise (e.g., whether coexisting AF and CCS independently contribute to unfavourable prognosis, whether AF (+)/CCS (−) patients have similar long-term mortality as AF (−)/CCS (+) patients, whether causes of death differ in relation to the diagnosis of AF and/or CCS).

The objectives of this study are to investigate the causes of the long-term mortality in patients referred for elective coronary angiography in relation to the diagnosis of AF and/or CCS, and second, to identify the factors that predispose to death in these patients.

## 2. Materials and Methods

### 2.1. Study Design

The Bialystok Coronary Project is a retrospective cohort study of consecutive patients with confirmed or suspected obstructive CCS conducted in the Department of Invasive Cardiology of the Medical University of Bialystok, Poland.

Patients were recruited between 2007 and 2016. In total, we screened 26,985 patients from Białystok, the largest city in north-eastern Poland. We excluded patients with acute coronary syndromes (ACS), Takotsubo cardiomyopathy, and a history of ischemic heart disease, as well as those referred for coronary angiography before heart valve surgery. Prior heart valve replacement was also an exclusion criterion. The final sample of the Bialystok Coronary Project consisted of 8288 patients referred for elective coronary angiography. Study details and outcomes have been presented previously [9].

We conducted a two-step follow-up examination. In the first stage, the general type of medication prescribed at discharge and planned revascularization treatment was evaluated. In the second stage, data on all-cause mortality was collected from the National Statistical Office in Poland. The exact collection date was 1 January 2019. The median duration of follow-up was 2029 (1283–3059) days. The records included information on the date and causes of death. The first aim of the present study was to investigate the relationship between long-term mortality and the diagnosis of AF and/or CCS. As shown in Figure 1, we divided our cohort into four subgroups, as follows: AF (+)/CCS (+), AF (+)/CCS (−), AF (−)/CCS (+), and AF (−)/CCS (−). Second, we investigated the predictors of the long-term all-cause and cardiovascular (CV) mortality in particular in the overall study population and the above-listed subgroups.

The study protocol conformed to the ethical guidelines of the Declaration of Helsinki and STROBE guidelines [14]. Additionally, it was approved by the local bioethics committee of the Medical University of Bialystok (Approval No. R-1-002/18/2019) and registered in the database of clinical studies www.clinicaltrials.gov (accessed on 23 October 2021) (Identifier: NCT04541498).

### 2.2. Study Parameters and Definitions

The CCS diagnosis was established according to the European guidelines in force at that time [15]. A significant stenosis of the coronary vessel (obstructive stenosis) was defined as stenosis of 50% or more of the diameter of the left main stem coronary artery or stenosis of 70% or more of the diameter of the rest of the arteries. We classified patients not fulfilling this criterion as CCS (−). In the group of patients with an obstructive coronary lesion, we classified CCS as single-, double-, or multi-vessel disease (MVD) defined as a triple-vessel disease and/or significant left main stem stenosis. A decision regarding optimal CCS management in our study participants (i.e., percutaneous coronary interventions (PCI), conservative management, or coronary bypass grafting surgery (CABG)) was performed by the attending international cardiologist and if required by the members of our Heart Team according to the guidelines current at the time of hospitalization.

We based diagnoses and AF classification on physician-assigned diagnoses in medical records corresponding to ICD-10-CM codes for AF at the hospital discharge or outpatient databases. The diagnoses were made based on the medical history, 24-h ECG monitoring, and standard 12-lead ECG performed on admission. AF was subclassified into paroxysmal, persistent, or permanent.

Left ventricular ejection fraction (LVEF) was assessed in the routine transthoracic echocardiography using the modified biplane Simpson’s method, following the European Society of Echocardiography recommendations [16]. The estimated glomerular filtration rate (eGFR) using the CKD-EPI formula and chronic kidney disease (CKD) was accessed according to the KDIGO 2012 Clinical Practice Guideline for the Evaluation and Management of Chronic Kidney Disease [17]. Obesity was defined as a body mass index ≥30 kg/m^2^. The diagnosis of other coexisting conditions was made based on medical history, physical examination results, and additional tests by the attending physician; it was not re-examined at the time of inclusion into the study.

The medications prescribed at discharge were divided into seven groups: acetylsalicylic acid (ASA), dual antiplatelet therapy (DAPT), vitamin K antagonists (VKAs), direct oral anticoagulants (DOACs), angiotensin-converting-enzyme inhibitors/angiotensin II receptor blockers (ACEI/ARB), beta-adrenergic antagonists (BB), and statins. Dual antiplatelet therapy (DAPT) was defined as taking acetylsalicylic acid (ASA) with clopidogrel, a platelet P2Y_12_ receptor inhibitor. The group of DOACs includes dabigatran, rivaroxaban, and apixaban.

### 2.3. Long-Term Mortality Data

The records collected from the National Statistical Office included information on the date and the causes of deaths recorded (codes in the International Classification of Diseases (ICD)—10th Revision). According to codes, we extracted the data for CV-related mortality (ICD-10 codes from I00 to I99).

### 2.4. Statistical Analysis

Data were collected and analysed using MS Excel (Microsoft, 2020, version 16.40). We used the Kolmogorov–Smirnov test to assess the distribution of continuous variables. None of the continuous variables followed the Gaussian distribution. Data were presented as medians (Me) and interquartile range (IQR) for not normally distributed continuous variables, and as the number (N) of cases and percentage (%) for categorical variables.

Statistical significance of differences between two groups were determined using the χ^2^ and Mann–Whitney U tests when appropriate. To compare multiple subgroups for non-normally distributed variables, we applied Kruskal–Wallis test with multiple pairwise comparisons using the Steel–Dwass–Critchlow–Fligner procedure, whereas for the comparison of categorical variables χ^2^ test was used.

The associations between parameters and mortality risk were estimated by Cox proportional hazard regression univariate and multivariate models. The multivariate analysis included variables with a *p* value < 0.1 in the univariate analysis. The results are presented as hazard ratio (HR) and 95% confidence intervals (CI). Kaplan–Meier curves were used for graphical assessment of time-dependent mortality according to the presence of AF and/or CCS. Multiple comparisons between groups were performed by the Dunn–Sidak method. For all analyses, we set the level of statistical significance at *p* < 0.05.

All analyses were performed using XL Stat (Addinsoft, 2020, version 2020.03.01, New York, NY, USA), Stata (StataCorp LLC, 2020, version 17, Lakeway Drive, TX, USA), and MS Excel (Microsoft, 2020, version 16.40, Redmond, WA, USA).

## 3. Results

### 3.1. Participant Characteristics

The final analyzed population consisted of 7367 patients (54% men, median (IQR) age: 65 (58–73) years), out of whom 1484 (20.1%) had AF and 2881 (39.1%) were diagnosed with obstructive CCS. During the median follow-up of 2029 (range 1283–3059) days, 1201 (16.3%) study participants died (Figure 1). Patients who died during the follow-up were older, more likely to be male gender, had lower body mass index and left ventricular ejection fraction, significantly more often presented with AF, obstructive CCS, advanced CAD (i.e., multi-vessel coronary artery disease and/or left main stem stenosis), diabetes, CKD and chronic obstructive pulmonary disease (COPD). Patients who died were less likely to be diagnosed with hypertension and hyperlipidemia but had higher values of CHA_2_DS_2_-VASc score (Table 1).

Comparing two CCS (+) subgroups, the subgroup with AF was significantly less likely to have a multi-vessel obstructive CCS and chronic total occlusion than the subgroup without AF (Table 2). The latter subgroup was less often diagnosed with COPD.

### 3.2. Mortality Analysis in the Subgroups

As shown in Figure 2, the all-cause death was highest in the AF (+)/CCS (+) subgroup, followed by AF (+)/CCS (−), AF (−)/CCS (+) and AF (−)/CCS (−) patients. The crude all-cause mortality rate was higher in AF (+)/CCS (−) vs. AF (−)/CCS (+) patients (21.9% (210/957) vs. 19.5% (459/2354); *p* < 0.01). Similar results were evident for CV mortality (Table 3 and Appendix A). CV deaths were more frequent than non-CV deaths in all subgroups except AF (−)/CCS (−) patients (Table 3).

CV mortality in the AF (+)/CCS (+) subgroup (70.1%, N = 136) was higher than in the AF (−)/CCS (+) patients (53.2% (N = 244; *p* < 0.001)). Coronary heart disease was a more common cause of death in the AF (+)/CCS (+) subgroup than in the AF (−)/CCS (+) subgroup (33.0% (*N = 64) vs. 21.8% (N = 100); *p* < 0.001), as well as in the AF (+)/CCS (−) subgroup compared to the AF (−)/CCS (−) one (27.6% (N = 58) vs. 14.8% (N = 50); *p* < 0.001). When comparing patients without CCS, more patients with AF than without AF died due to intracerebral haemorrhage (5.7% (N = 12) vs. 1.5% (N = 5); *p* = 0.005), see Table 3.

### 3.3. Predictors of Mortality

Significant predictors found on univariate analysis are presented in Appendix A. Variables with a *p* value <0.1 in the univariate analysis were incorporated in the multivariate analysis. Figure 3 shows the predictors of all-cause and CV mortality in multivariate Cox proportional hazards models: AF and obstructive CCS together with increasing age, male gender, chronic heart failure, chronic obstructive pulmonary disease, diabetes mellitus, and lower values of red blood cells were independent predictors of both all-cause and CV mortality. Chronic kidney disease increased, but statin therapy decreased, the risk of all-cause mortality.

The presence of obstructive CCS increased all-cause and CV mortality risk by 2-fold and 3-fold, respectively. As shown in Figure 4, other independent predictors of total mortality in the AF subgroups were increasing age, chronic heart failure, COPD, and lower values of red blood cells, whereas male sex, chronic heart failure, diabetes mellitus, and lower values of red blood cells were associated with CV mortality. Increasing age was found to be a significant predictor of all-cause mortality in all subgroups, but was associated with CV mortality only in the AF (−)/CCS (+) and AF (−)/CCS (−) patients (Table 4). Male sex was associated with CV mortality in all subgroups and with all-cause mortality in all subgroups except the AF (+)/CCS (+) patients. Baseline diagnosis of chronic heart failure predicted both all-cause and CV mortality in all subgroups, especially in AF (−)/CCS (−) patients.

## 4. Discussion

In this large analysis from the Bialystok Coronary Project, our principal findings are as follows: (I) both AF and obstructive CCS were strong and independent predictors of the long-term all-cause and CV mortality; (II) mortality of AF patients without CCS was at least as high as non-AF patients with CCS; and (III) CV deaths were more frequent than non-CV deaths in AF patients with CCS compared to those with either AF or CCS alone. Our study not only highlights the fact that the diagnosis of AF remains a strong predictor of long-term mortality, but also clearly demonstrates that coexistence of AF with obstructive CCS further reduces the survival.

Specific causes of death are frequently not reported in studies exploring long-term mortality in AF patients. Similar to our data, Lee et al. found that among 15,411 AF patients from the Korean registry, CV mortality was more frequent than cancer-related mortality [18]. Additionally, AF patients had a 4-fold increased risk of all-cause mortality compared with the general population. In contrast to our study cohort, Lee et al. found that cerebral infarction (but not coronary heart disease) was the most common cause of death [18]. In Europe, Fauchier et al. analysed patients diagnosed with AF in four hospitals, and demonstrated that the majority of deaths were of CV origin [19], most commonly heart failure (29%), infection (18%), and cancer (12%), while fatal stroke or fatal bleeding each accounted for 7% of all deaths. These findings are concordant with our results in the overall population of the Białystok Coronary Project, with CV deaths more frequent than non-CV deaths in all subgroups except for the AF (−)/CCS (−) patients.

An increased short- and long-term mortality in ACS patients with coexisting AF remains a well-known phenomenon [7,20,21,22,23]. We hypothesize that pre-existing AF in ACS patients may be a marker of prior myocardial disease, while new-onset AF may be associated with more extensive myocardial injury in the course of ACS. Nonetheless, data from ACS studies do not necessarily apply to CCS patients. Of note, ACS patients were excluded from the present study.

Our findings correspond with the results of a Spanish study including 17,100 patients aged at least 50 years with known or suspected CCS who underwent exercise electrocardiography (N = 11,911) or exercise echocardiography (N = 5189) [24]. The highest long-term mortality in patients with AF and a positive stress test result when compared with other subgroups. In addition, the diagnosis of AF remained an independent predictor of all-cause mortality, but not of nonfatal myocardial infarction or coronary revascularization [24]. In an Austrian single-centre registry including 1434 patients with CCS and 1456 patients with ACS, patients undergoing elective or urgent coronary revascularization and suffering from AF had a 2-fold increased adjusted relative risk of death after a mean follow-up of 4.8 years [25]. Similar to our data, CKD, CCS, and diabetes were independent predictors of 1-year all-cause mortality in patients with both AF and chronic heart failure with reduced ejection fraction [26].

Another main finding of our work is that patients with AF but without obstructive CCS have a reduced survival when compared with those with obstructive CCS but without concomitant AF. In a cohort of patients referred for exercise stress testing for myocardial ischemia, Bouzas-Mosquera et al. demonstrated a higher long-term all-cause mortality in AF patients with negative stress testing compared to patients without AF [24]. However, patients with AF have coexisting obstructive CCS more often than those with sinus rhythm [27]. Given that the prevalence of obstructive CCS in patients with AF may be as high as 46.5%, it is possible that at the time of inclusion to the study the lesions in their epicardial coronary arteries were not yet so advanced so as to be considered significant at coronary angiography [6,28]. In addition, coronary atherosclerosis tends to progress over time and the co-occurrence of AF and obstructive CCS worsens the patients’ prognosis even when they are carefully treated [7,28].

We observed that AF (+)/CCS (−) vs. AF (−)/CCS (+) patients had a higher proportion of deaths due to intracerebral haemorrhage. This may be associated with an under-use of DOACs, which are not refundable in Poland, unlike the VKAs. A large study from various European countries from 2011–2016—overlapping with our study period—showed that out of patients taking oral anticoagulants, 67% were on VKAs and only 33% on DOACs [29].

Our study has limitations. First, our follow-up includes only mortality, there are no data on the condition of patients’ health and nonfatal clinical events after inclusion into the study. Second, our findings were obtained in a retrospective single-center study and should be verified in a prospective multicenter study. Third, our observations are restricted to elective patients as those with ACS were excluded from the study. Fourth, fractional flow reserve measurements were not performed on regular basis in our study participants and therefore assessment of the significance of coronary stenoses might have been inaccurate in some of our patients. Fifth, we were not able to obtain reliable data on AF ablation procedures, smoking status, and diabetes therapy in our study participants which may be a confounder in our analysis. Sixth, due to the high count of garbage codes in total mortality in Poland, case-specific mortality is likely to be underestimated. Finally, many of AF patients in our study also suffered from heart failure and vice versa. Additionally, heart failure was a common cause of mortality in our study. These facts might affect our findings.

## 5. Conclusions

Among patients referred for elective coronary angiography, both AF and obstructive CCS are strong and independent predictors of the long-term all-cause and CV mortality. Mortality of AF without CCS was at least as high as non-AF patients with CCS. CV deaths were more frequent than non-CV deaths in AF patients with CCS compared to those with either AF or CCS alone. Therefore, we recommend a careful clinical follow-up of AF patients, with a particular emphasis on stroke prevention and modification of CV risk factors.

## Figures and Tables

**Figure 1 jcm-10-04949-f001:**
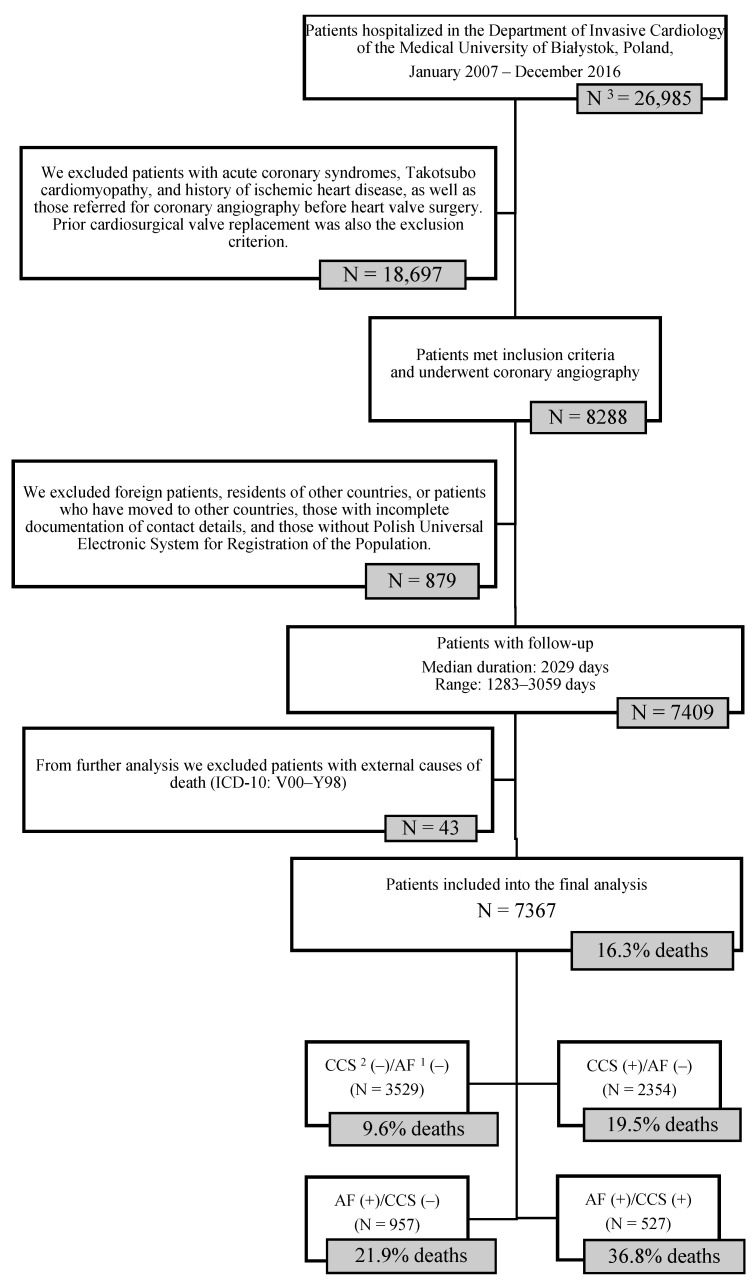
Selection of the study population. ^1^ AF, atrial fibrillation; ^2^ CCS, chronic coronary syndrome; ^3^ N, number.

**Figure 2 jcm-10-04949-f002:**
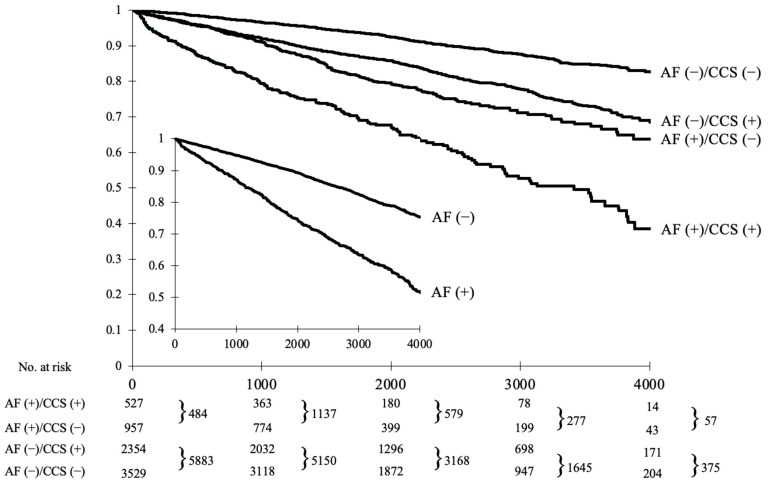
Kaplan–Meier survival analysis of all-cause mortality in relation of the diagnosis of atrial fibrillation and obstructive chronic coronary syndrome (large graph). The inner graph represents the comparison between patients with and without atrial fibrillation independently of the diagnosis of obstructive chronic coronary syndrome. All differences between curves are statistically significant (adjusted *p* values < 0.01 for all tests). AF, atrial fibrillation; CCS, chronic coronary syndrome.

**Figure 3 jcm-10-04949-f003:**
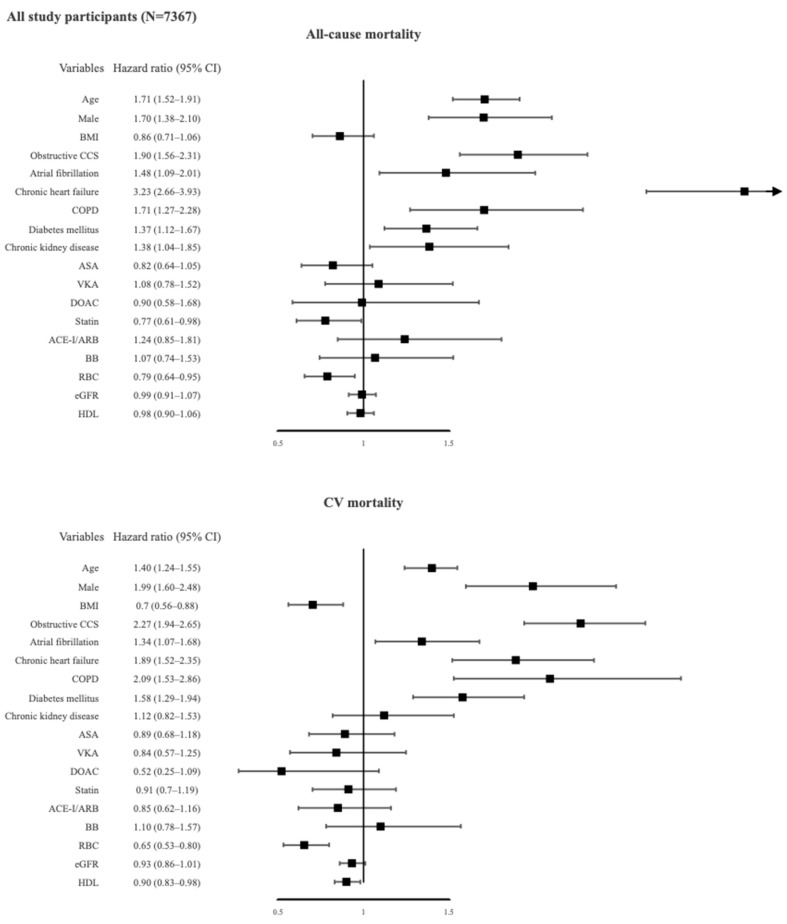
Predictors of all-cause and cardiovascular mortality in all study participants using multivariate analysis. ACEI, angiotensin-converting-enzyme inhibitor; ARB, angiotensin receptor blocker; ASA, acetylsalicylic acid; BB, beta adrenergic receptor antagonist; BMI, body mass index; CCS, chronic coronary syndrome; COPD, chronic obstructive pulmonary disease; CV, cardiovascular; DOAC, direct oral anticoagulant; eGFR, estimated glomerular filtration rate; HDL, high-density lipoprotein cholesterol; RBC, red blood cells; VKA, vitamin K antagonist.

**Figure 4 jcm-10-04949-f004:**
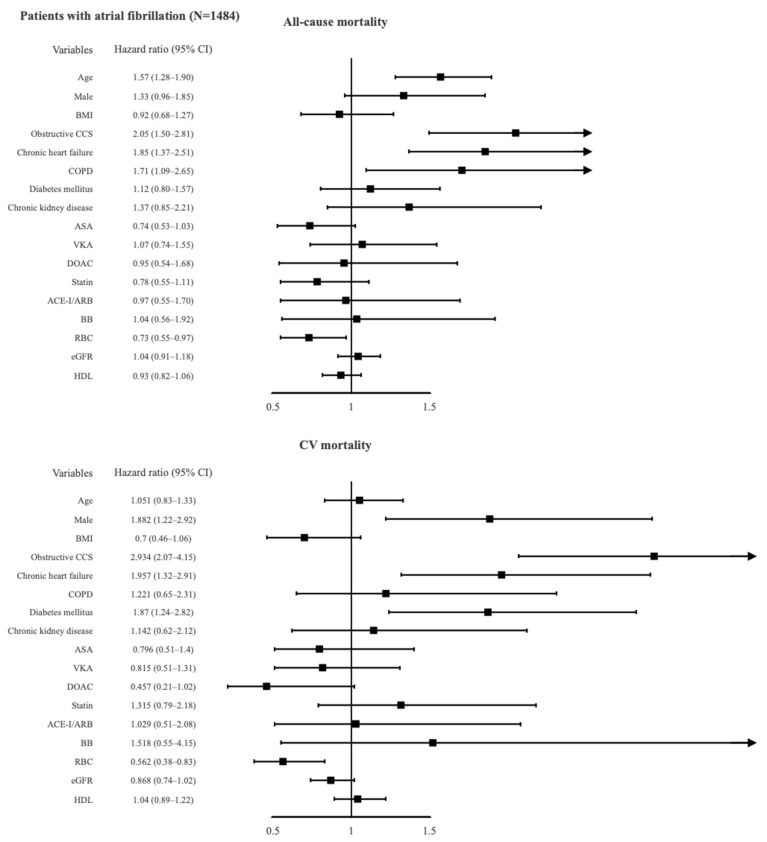
Predictors of all-cause and cardiovascular mortality in patients with atrial fibrillation using multivariate analysis. ACEI, angiotensin-converting-enzyme inhibitor; ARB, angiotensin receptor blocker; ASA, acetylsalicylic acid; BB, beta adrenergic receptor antagonist; BMI, body mass index; CCS, chronic coronary syndrome; COPD, chronic obstructive pulmonary disease; CV, cardiovascular; DOAC, direct oral anticoagulant; eGFR, estimated glomerular filtration rate; HDL, high-density lipoprotein cholesterol; RBC, red blood cells; VKA, vitamin K antagonist.

**Table 1 jcm-10-04949-t001:** Baseline characteristics in the whole cohort, and alive and dead participants at follow-up.

	All Study Participants (N = 7367)	Alive Participants (N = 6166)	Dead Participants (N = 1201)	*p* Value for the Comparison between Alive and Dead Study Participants
Age, years; Me ^13^ (IQR ^11^)	65 (58–73)	64 (57–72)	71 (62–77)	<0.001
Male; % (N ^14^)	54 (3978)	51.4 (3168)	67.4 (810)	<0.001
BMI ^5^; Me (IQR)	29 (26–32)	29 (26–32)	28 (25–31)	<0.001
Obesity; % (N)	35.0 (2582)	36.1 (2225)	29.7 (357)	<0.001
Atrial fibrillation; % (N)	20.1 (1484)	17.5 (1080)	33.6 (404)	<0.001
Paroxysmal atrial fibrillation; % (N)	8.9 (656)	8.5 (525)	10.9 (131)	0.02
Persistent atrial fibrillation; % (N)	8.8 (649)	6.8 (417)	19.3 (232)	
Permanent atrial fibrillation; % (N)	2.4 (179)	2.2 (138)	3.4 (41)	
Hypertension; % (N)	82.8 (6103)	83.3 (5134)	80.7 (969)	0.03
Diabetes mellitus; % (N)	25.5 (1878)	24.2 (1492)	32.1 (386)	<0.001
Hyperlipidaemia; % (N)	88.6 (6526)	89.3 (5505)	85.0 (1021)	<0.001
Low-density lipoprotein cholesterol, mg/dL; Me (IQR)	100 (78–128)	100 (79–128)	100 (77–127)	0.35
High-density lipoprotein cholesterol, mg/dL; Me (IQR)	46 (39–56)	47 (39–56)	44 (37–53)	<0.001
Chronic heart failure; % (N)	18.3 (1345)	14.3 (884)	38.4 (461)	<0.001
LVEF ^12^, %; Me (IQR)	55 (43–60)	55 (48–60)	45 (30–55)	<0.001
Chronic kidney disease; % (N)	20 (1471)	16.6 (1021)	37.5(450)	<0.001
eGFR ^10^, mL/min/1.73 m^2^; Me (IQR)	79 (65–91)	81 (68–92)	71 (54–86)	<0.001
COPD ^7^, % (N)	4.5 (335)	3.4 (211)	10.3 (124)	<0.001
Obstructive CCS ^6^; % (N)	39.1 (2881)	36.1 (2228)	54.4 (653)	<0.001
Single-vessel CCS; % (N)	17.2 (1270)	16.1 (993)	23.1 (277)	<0.001
Double-vessel CCS; % (N)	9.6 (710)	9.0 (555)	12.9 (155)	
Multi-vessel CCS; % (N)	21.9 (1611)	20.0 (1235)	31.3 (376)	
Significant stenosis; % (N)				
Left main	3 (220)	2.7 (166)	4.5 (54)	<0.001
Left anterior descending artery	24.9 (1837)	23.2 (1428)	34.1 (409)	<0.001
Diagonal artery	8.9 (654)	8.1 (501)	12.7 (153)	<0.001
Circumflex artery	13.3 (976)	12.1 (747)	19.1 (229)	<0.001
Left marginal artery	8.6 (634)	8.0 (492)	11.2 (142)	<0.001
Right coronary artery	19.1 (1405)	16.8 (1037)	30.6 (368)	<0.001
Chronic total occlusion; % (N)	8.1 (596)	7.3 (451)	12.1 (145)	<0.001
Left main	0.4 (32)	0.4 (23)	0.7 (9)	0.61
Left anterior descending artery	5.6 (414)	5.0 (306)	9.0 (108)	0.03
Diagonal artery	1.8 (133)	1.6 (99)	2.8 (34)	0.51
Circumflex artery	3.0 (222)	2.5 (154)	5.7 (68)	0.004
Left marginal artery	2.2 (163)	2.0 (123)	3.3 (40)	0.45
Right coronary artery	5.3 (391)	4.7 (289)	8.5 (102)	0.96
Patients treated with sucessful PCI, % (N)				
Left main	0.6 (41)	0.4 (27)	1.2 (14)	0.52
Left anterior descending artery	13.1 (965)	12.6 (779)	15.5 (186)	0.02
Diagonal artery	3.9 (285)	3.5 (218)	5.6 (67)	0.91
Circumflex artery	6.5 (479)	6.2 (380)	8.2 (99)	0.44
Left marginal artery	3.3 (244)	3.3 (205)	3.2 (39)	0.94
Right coronary artery	8.5 (628)	7.9 (489)	11.6 (139)	0.02
Unsuccessful PCI	0.9 (64)	0.9 (55)	0.7 (9)	0.64
Medication prescribed at discharge; % (N)				
ASA ^3^	81.1 (5976)	81.8 (5046)	77.4 (930)	<0.001
DAPT ^8^	21.7 (1599)	21.1 (1300)	24.9 (299)	0.003
DOAC ^9^	4.3 (316)	4.5 (277)	3.3 (39)	0.051
VKA ^15^	12.4 (914)	10.4 (643)	22.6 (271)	<0.001
ACEI ^1^/ARB ^2^	87.4 (6438)	86.8 (5353)	90.4 (1085)	<0.001
BB ^4^	89.5 (6595)	89.0 (5486)	92.4 (1109)	<0.001
Statin	83.9 (6184)	84.5 (5208)	81.3 (976)	0.006
CHA_2_DS_2_-VASc score; Me (IQR)	3 (2–4)	3 (2–4)	4 (3–5)	<0.001

^1^ ACEI, angiotensin-converting-enzyme inhibitor; ^2^ ARB, angiotensin receptor blocker; ^3^ ASA, acetylsalicylic acid; ^4^ BB, beta adrenergic receptor antagonist; ^5^ BMI, body mass index; ^6^ CCS, chronic coronary syndrome; ^7^ COPD, chronic obstructive pulmonary disease; ^8^ DAPT, dual antiplatelet therapy; ^9^ DOAC, direct oral anticoagulant; ^10^ eGFR, estimated glomerular filtration rate; ^11^ IQR, interquartile range; ^12^ LVEF, left ventricular ejection fraction; ^13^ Me, median; ^14^ N, number; ^15^ VKA, vitamin K antagonist.

**Table 2 jcm-10-04949-t002:** Differences in the characteristic of study participants according to the presence of atrial fibrillation and obstructive chronic coronary syndrome.

	AF (+)/CCS (+) ^a^ (N = 527)	AF (+)/CCS (−) ^b^ (N = 957)	AF (−)/CCS (+) ^c^ (N = 2354)	AF (−)/CCS (−) ^d^ (N = 3529)	*p* Value for the Comparison among Sub-Groups
Age, years; Me (IQR)	71 (66–77)	68 (60–74)	66 (59–73)	63 (56–71)	<0.001
Male; % (N)	68.7 (362)	57.2 (547)	66.7 (1570)	42.5 (1499)	<0.001 ^1^
Obesity; % (N)	36.1 (190)	42.3 (405)	31.7 (747)	35.1 (1240)	<0.001 ^2^
MVD; % (N)	45.5 (241)	N/A	58.2 (1370)	N/A	<0.001
Chronic total occlusion; % (N)	15.9 (84)	N/A	21.8 (512)	N/A	<0.001
Hypertension; % (N)	83.9 (442)	79.6 (762)	86.8 (2042)	81 (2857)	<0.001 ^3^
Diabetes mellitus; % (N)	32.1 (169)	26.3 (252)	29.9 (704)	21.3 (753)	<0.001 ^4^
Hyperlipidaemia; % (N)	89.9 (474)	79.9 (765)	96.2 (2265)	85.6 (3022)	<0.001
Chronic heart failure; % (N)	42.1 (222)	39.5 (378)	14.5 (341)	11.4 (404)	<0.001 ^5^
Chronic kidney disease; % (N)	41.0 (216)	28.4 (272)	20.2 (475)	14. (508)	<0.001
COPD, % (N)	7.2 (38)	6.6 (63)	4.0 (94)	4.0 (140)	<0.001 ^6^
Conservative management, % (N)	13.3 (70)	N/A	11.7 (275)	N/A	<0.001
Patients sucessfully treated with PCI, % (N)	46.2 (233)	N/A	50.1 (1180)	N/A	
Unsuccessful PCI, %, (N)	2.7 (14)	N/A	2.1 (50)	N/A	
Patients qualified for CABG, % (N)	39.9 (210)	N/A	36.1 (849)	N/A	
DOACs prescribed at discharge, % (N)	15.7 (83)	22.3 (213)	0.2 (5)	0.4 (15)	<0.001 ^7^
VKAs prescribed at discharge, % (N)	56.4 (297)	57.3 (548)	0.8 (21)	1.3 (48)	<0.001 ^6^
CHA_2_DS_2_ -VASc score; Me (IQR)	5 (4–5)	4 (3–5)	4 (3–5)	3 (2–4)	<0.001

^1^ no significant differences between the ^a^ vs. ^c^ groups. ^2^ no significant differences between the ^a^ vs. ^d^ and ^a^ vs. ^c^ groups. ^3^ no significant differences between the ^a/b^ vs. ^d^ and ^b/c^ vs. ^a^ groups. ^4^ no significant differences between the ^b/c^ vs. ^a^ and ^b^ vs. ^c^ groups. ^5^ no significant differences between the ^a^ vs. ^b^ groups. ^6^ no significant differences between the ^a^ vs. ^b^ and ^c^ vs. ^d^ groups. ^7^ no significant differences between the ^c^ vs. ^d^ groups. AF, atrial fibrillation; CABG; coronary artery bypass grafting; CCS; chronic coronary syndrome; COPD, chronic obstructive pulmonary disease; IQR, interquartile range; Me, median; MVD, multivessel disease (CCS with significant triple-vessel and/or left main stem stenosis); N/A, not applicable; N, number.

**Table 3 jcm-10-04949-t003:** Causes of deaths in the study participants.

All Deaths, % (N)		Total 16.3 (1201)	AF (+)/CCS (+) ^a^ 36.8 (194)	AF (+)/CCS (−) ^b^ 21.9 (210)	AF (−)/CCS (+) ^c^ 19.5 (459)	AF (−)/CCS (−) ^d^ 9.6 (338)	*p* Value for the Comparison among Subgroups
Cardiovascular deaths, % * (N)		53.5 (643)	70.1 (136)	58.6 (123)	53.2 (244)	41.4 (140)	<0.001 ^1^
Non-cardiovascular deaths, % (N)		46.5 (558)	20.9 (58)	52.4 (87)	46.8 (215)	58.6 (198)	<0.001 ^1^
	Neoplasm deaths, % (N)	25.6 (307)	14.9 (29)	20 (42)	27.9 (128)	32 (108)	<0.001 ^2^
	Other deaths, % (N)	20.9 (251)	14.9 (29)	21.4 (45)	19 (87)	26.6 (90)	0.01 ^3^
Leading causes of death							
Coronary heart disease, % (N)		22.6 (272)	33 (64)	27.6 (58)	21.8 (100)	14.8 (50)	<0.001 ^4^
Lung cancer, % (N)		7.2 (87)	4.1 (8)	3.3 (7)	8.3 (38)	10.1 (34)	0.006 ^5^
Cardiomyopathy, % (N)		3.2 (38)	2.1 (4)	4.3 (9)	1.1 (5)	5.9 (20)	0.001 ^6^
Cerebral infarction, % (N)		5.1 (61)	6.7 (13)	7.6 (16)	4.8 (22)	3 (10)	0.07
Myocardial infarction, % (N)		5 (60)	6.2 (12)	1.9 (4)	6.3 (29)	4.4 (15)	0.08
Instantaneous death, % (N)		4.9 (59)	4.1 (8)	6.2 (13)	4.6 (21)	5 (17)	0.77
Heart failure, % (N)		4.5 (54)	6.7 (13)	4.3 (9)	4.1 (19)	3.8 (13)	0.44
Pneumonia, % (N)		3 (36)	0.5 (1)	3.8 (8)	3.1 (14)	3.8 (13)	0.14
Intracerebral haemorrhage, % (N)		2.3 (28)	2.1 (4)	5.7 (12)	1.5 (7)	1.5 (5)	0.005 ^7^
Prostate cancer, % (N)		2.2 (27)	1 (2)	3.3 (7)	2.6 (12)	1.8 (6)	0.38
Colon cancer, % (N)		2 (24)	2.6 (5)	2.4 (5)	2 (9)	1.5 (5)	0.81
COPD, % (N)		1.8 (22)	0.5 (1)	1.4 (3)	1.3 (6)	3.6 (12)	0.06
Hypertensive heart disease, % (N)		1.7 (20)	0.5 (1)	0.5 (1)	2 (9)	2.7 (9)	0.13
Diabetes mellitus, % (N)		1.6 (19)	3.1 (6)	0.5 (1)	1.7 (8)	1.2 (4)	0.18
Gastric cancer, % (N)		1.5 (18)	1 (2)	0 (0)	1.7 (8)	2.4 (8)	0.15
Atherosclerosis, % (N)		1.3 (16)	2.6 (5)	1 (2)	1.7 (8)	0.3 (1)	0.12
Pancreatic cancer, % (N)		1.2 (14)	0 (0)	0.5 (1)	1.5 (7)	1.8 (6)	0.2
Brain tumour, % (N)		1.2 (14)	0.5 (1)	1 (2)	1.1 (5)	1.8 (6)	0.6
Intestinal ischemia, % (N)		1.2 (14)	1 (2)	1 (2)	1.1 (5)	1.5 (5)	0.94
Aortic valve disorders, % (N)		1.1 (13)	2.6 (5)	0.5 (1)	1.3 (6)	0.3 (1)	0.07
Pulmonary hypertension, % (N)		1 (12)	0.5 (1)	1 (2)	0.9 (4)	1.5 (5)	0.72
Other, % (N)		23.6 (283)	17 (33)	21.9 (46)	24.2 (111)	27.5 (93)	<0.001

* percentage of deaths. ^1^ no significant differences between the ^a/c^ vs. ^b^ groups. ^2^ no significant differences between the ^a/c^ vs. ^b^ and ^c^ vs. ^d^ groups. ^3^ no significant differences between the ^a/c/d^ vs. ^b^ and ^a^ vs. ^c^ groups. ^4^ no significant differences between the ^a/c^ vs. ^b^ and ^d^ vs. ^c^ groups. ^5^ significant differences between the ^d^ vs. ^b^ groups. ^6^ significant differences between the ^b/d^ vs. ^c^ groups. ^7^ significant differences between the ^c/d^ vs. ^b^ groups. AF, atrial fibrillation; CCS; chronic coronary syndrome; COPD, chronic obstructive pulmonary disease; N, number.

**Table 4 jcm-10-04949-t004:** Predictors of all-cause and cardiovascular mortality in the subgroups according to the presence of atrial fibrillation and obstructive chronic coronary syndromes. Multivariate analysis using Cox proportional hazards models. Results are shown as hazard ratios with 95% confidence intervals and corresponding *p* values.

Variables		AF (+)/CCS (+) ^a^ (N = 527)	AF (+)/CCS (−) ^b^ (N = 957)	AF (−)/CCS (+) ^c^ (N = 2354)	AF (−)/CCS (−) ^d^ (N = 3529)	*p* Value for the Comparison among Subgroups
Age, (for a 10-year increase)	All-cause mortality	1.68 (1.24–2.24) *p* < 0.001	1.55 (1.17–2.04) *p* < 0.001	1.79 (1.51–2.14) *p* < 0.001	1.64 (1.32–2.06) *p* < 0.001	*p* = 0.67
	CV mortality	1.21 (0.83–1.77) *p* = 0.32	0.95 (0.69–1.29) *p* = 0.73	1.38 (1.16–1.64) *p* < 0.001	1.64 (1.37–1.97) *p* < 0.001	*p* = 0.04 ^1^
Male	All-cause mortality	0.92 (0.58–1.44) *p* = 0.71	1.93 (1.18–3.16) *p* = 0.01	1.45 (1.03–2.05) *p* = 0.03	2.76 (1.76–4.33) *p* < 0.001	*p* = 0.03 ^2^
	CV mortality	2.44 (1.13–5.30) *p* = 0.02	1.77 (1.03–3.07) *p* = 0.04	1.52 (1.05–2.21) *p* = 0.03	2.57 (1.81–3.64) *p* < 0.001	*p* = 0.045 ^3^
Chronic heart failure	All-cause mortality	1.65 (1.08–2.53) *p* = 0.02	2.32 (1.47–3.65) *p* < 0.001	3.64 (2.69–4.92) *p* < 0.001	6.55 (4.33–9.91) *p* < 0.001	*p* = 0.024 ^4^
	CV mortality	2.23 (1.18–4.24) *p* = 0.01	1.99 (1.17–3.39) *p* = 0.01	1.83 (1.27–2.63) *p* < 0.001	1.93 (1.30–2.87) *p* < 0.001	*p* = 0.77
Diabetes mellitus	All-cause mortality	1.31 (0.82–2.07) *p* = 0.26	0.94 (0.57–1.56) *p* = 0.82	1.44 (1.07–1.95) *p* = 0.02	1.59 (1.03–2.46) *p* = 0.04	*p* = 0.43
	CV mortality	2.86 (1.47–5.55) *p* < 0.001	1.75 (1.00–3.07) *p* = 0.05	1.33 (0.96–1.84) *p* = 0.09	1.76 (1.24–2.51) *p* < 0.001	*p* = 0.04 ^5^
Statin	All-cause mortality	0.90 (0.51–1.60) *p* = 0.73	0.77 (0.49–1.22) *p* = 0.27	0.81 (0.49–1.33) *p* = 0.4	0.87 (0.53–1.43) *p* = 0.58	*p* = 0.81
	CV mortality	0.88 (0.37–2.1) *p* = 0.78	1.52 (0.81–2.84) *p* = 0.19	1.15 (0.63–2.09) *p* = 0.65	0.62 (0.43–0.91) *p* = 0.01	*p* = 0.02 ^6^
ACEI/ARB at discharge	All-cause mortality	1.51 (0.59–3.84) *p* = 0.39	0.57 (0.27–1.19) *p* = 0.14	0.93 (0.53–1.64) *p* = 0.8	3.27 (1.01–10.59) *p* = 0.05	*p* = 0.01 ^7^
	CV mortality	2.60 (0.73–9.34) *p* = 0.14	0.86 (0.36–2.08) *p* = 0.74	0.95 (0.53–1.68) *p* = 0.85	0.72 (0.46–1.13) *p* = 0.15	*p* = 0.42
RBC, (for a 10^6^/mm^3^ increase)	All-cause mortality	0.75 (0.52–1.10) *p* = 0.14	0.68 (0.44–1.05) *p* = 0.08	0.77 (0.56–1.05) *p* = 0.1	1.27 (0.84–1.91) *p* = 0.26	*p* = 0.04 ^7^
	CV mortality	0.45 (0.25–0.82) *p* = 0.01	0.63 (0.36–1.11) *p* = 0.11	0.70 (0.50–0.99) *p* = 0.05	0.62 (0.43–0.88) *p* = 0.01	*p* = 0.55
HDL (for a 10 mg/dL increase)	All-cause mortality	1.01 (0.83–1.22) *p* = 0.94	0.88 (0.72–1.06) *p* = 0.18	0.92 (0.81–1.05) *p* = 0.24	1.13 (0.9–1.31) *p* = 0.11	*p* = 0.048 ^8^
	CV mortality	0.90 (0.68–1.17) *p* = 0.42	1.16 (0.95–1.42) *p* = 0.15	0.85 (0.74–0.98) *p* = 0.02	0.89 (0.78–1.01) *p* = 0.07	*p* = 0.03 ^1^

^1^ significant difference between the ^b^ vs. ^c/d^ groups. ^2^ significant differences between the ^d^ vs. ^a/c^ and ^a^ vs. ^b^ groups. ^3^ significant differences between the ^b^ vs. ^c^ groups. ^4^ no significant differences between the ^b^ vs. ^a/c^ groups. ^5^ significant differences between the ^a^ vs. ^c^ groups. ^6^ significant differences between the ^a^ vs. ^b^ groups. ^7^ significant differences between the ^b^ vs. ^d^ groups. ^8^ significant differences between the ^d^ vs. ^b/c^ groups. ACEI, angiotensin-converting-enzyme inhibitor; ARB, angiotensin receptor blocker; CV, cardiovascular; HDL, high-density lipoprotein cholesterol; RBC, red blood cells.

## Data Availability

The data presented in this study are available on request from the corresponding author.

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
