# Peer review of "Predicting Mortality in Patients with Atrial Fibrillation and Obstructive Chronic Coronary Syndrome: The Bialystok Coronary Project"

_jcm, 2021, doi:10.3390/jcm10214949_

Round 1

Reviewer 1 Report

The study aims to analyse the relation between atrial fibrillation and chronic coronary artery disease.

The presentation is clean and balanced.

I only have a few comments.

Major comments:

  1. Can the authors provide a sub-analysis to focus on success and fialure of PCIs performed in the described cohort?
  2. Also, can the authors present the rate of chronic total occlusion in CCS group? Possibly a sub analysis by logistic regression could further improve our understanding which sub-group is in clear dange - revascularized, non-revascularized, CTO or only chronic stenoses?
  3. Heart failure is a leading cause for all cause mortality as presented. Many of AF patients are also suffer from heart failure and vice versa. Can the authors comments on that. (Or possibly assess the relation between these factors?)

Minor comments:

  1. A structured abstract is recommended.
  2. I would rephrase the conclusion part, or extend it in light how this research could further improve clinical practice.
  3. What is the rate of AF invasive treatment?
  4. Conclusions come after limitations.

Author Response

Reviewer #1

We are grateful for the time and effort invested in reviewing our manuscript and for the many thoughtful comments, kind words, and suggestions from the Reviewer. We have considered these comments carefully and feel that we have fully addressed the reviewer’s concerns. We have replied to each comment below and edited the manuscript accordingly. We believe that this input has improved this manuscript. Complying with all the suggestions, we hope that you will find the manuscript suitable for publication in JCM.

Major comments:

  1. Can the authors provide a sub-analysis to focus on success and failure of PCIs performed in the described cohort?

Thank you for this suggestion. We performed such analyses and added their results to both Table 1 and Table 2. They now include information on all PCIs performed, with a breakdown by the individual arteries, as well as all failed percutaneous interventions.

  1. Also, can the authors present the rate of chronic total occlusion in CCS group? Possibly a sub analysis by logistic regression could further improve our understanding which sub-group is in clear dange - revascularized, non-revascularized, CTO or only chronic stenoses?

Thank you for this question. We have added data regarding CTO and in the comparison of deceased and living patients, there are no significant differences regarding CTO regardless of the type of artery. Unfortunately, building regression models including CTOs of individual arteries was unsuccessful. Additionally, the topic of CTO is beyond the focus of our analysis, as the work was more focused on atrial fibrillation, but we will keep this suggestion in mind in future work designed more adequately.

  1. Heart failure is a leading cause for all cause mortality as presented. Many of AF patients are also suffer from heart failure and vice versa. Can the authors comments on that. (Or possibly assess the relation between these factors?

We agree with the Reviewer. There, we added to the study limitations the following sentences:

“Finally, many of AF patients in our study also suffered from heart failure and vice versa. Additionally, heart failure was a common cause of mortality in our study. These facts might affect our findings.”

Minor comments:

  1. A structured abstract is recommended.

According to the instructions for authors in JCM, the form of the abstract should only follow the style of structured abstract, but in its final form be without headings:

“The abstract should be a single paragraph and should follow the style of structured abstracts, but without headings: 1) Background: Place the question addressed in a broad context and highlight the purpose of the study; 2) Methods: Describe briefly the main methods or treatments applied. Include any relevant preregistration numbers, and species and strains of any animals used. 3) Results: Summarize the article's main findings; and 4) Conclusion: Indicate the main conclusions or interpretations”

We believe that our abstract is in accordance with the above guidelines.

  1. I would rephrase the conclusion part, or extend it in light how this research could further improve clinical practice.

We thank the Reviewer for this valuable suggestion and in accordance with it the following sentence is now added into the Conclusions section:

“Therefore, we recommend a careful clinical follow-up of AF patients, with a particular emphasis on stroke prevention and modification of CV risk factors.”

  1. What is the rate of AF invasive treatment?

This is a great question, unfortunately we do not have data regarding the AF invasive treatment in our study group. This fact was added to the revised study limitations section. We will want to expand our database with this information in the future and use it in subsequent analyses.

  1. Conclusions come after limitations.

We agree with this, as it is in line with the JCM instructions for authors, therefore conclusions of our study come directly after limitations, which are located in the last paragraph of the discussion.

Reviewer 2 Report

This study investigated the causes of the long-term mortality in patients referred for elective coronary angiography in relation to the diagnosis of AF and/or CCS, and second, to identify the factors that predispose to death in these patients. The manuscript is well written and structured, the information it is provided is clear and concise. The main limitation of the study is the sample size. No data have been supplied to verify if the number of patients enrolled is enough to detect a difference among the several study groups. For example, sudden cardiac death has been reported to be high in patients with coronary artery disease and AF as well as in patients with heart failure and AF. In Table 3, both sudden cardiac death and heart failure did not differ among the four study groups. Please clarify this crucial point.

Please define the length of the follow-up. So far it could be deduced only from Figure 1.

Author Response

Reviewer #2

This study investigated the causes of the long-term mortality in patients referred for elective coronary angiography in relation to the diagnosis of AF and/or CCS, and second, to identify the factors that predispose to death in these patients. The manuscript is well written and structured, the information it is provided is clear and concise. The main limitation of the study is the sample size. No data have been supplied to verify if the number of patients enrolled is enough to detect a difference among the several study groups. For example, sudden cardiac death has been reported to be high in patients with coronary artery disease and AF as well as in patients with heart failure and AF. In Table 3, both sudden cardiac death and heart failure did not differ among the four study groups. Please clarify this crucial point.

Please define the length of the follow-up. So far it could be deduced only from Figure 1.

We are grateful for the time and effort invested in reviewing our manuscript and for the many thoughtful comments, kind words, and suggestions from the Reviewer. We have considered these comments carefully and feel that we have fully addressed the reviewer’s concerns. We have replied to each comment below and edited the manuscript accordingly. We believe that this input has improved this manuscript. Complying with all the suggestions, we hope that you will find the manuscript suitable for publication in JCM.

As for the matter of study sample size. This is a single-center study, which in this case is an apparent limitation of our work that we mention in the manuscript, but we believe that the 10-year period of patient enrollment for the analysis compensates for the final number of patients. Nevertheless, as we mentioned in the study limitations section, further multicenter, prospective studies are necessary to confirm our findings.

Unfortunately, due to the high count of garbage codes in total mortality in Poland (which includes heart failure), case-specific mortality is likely to be underestimated, which is why we focused on all-cause and cardiovascular mortality rather than the case-specific one. This limitation is now added to the manuscript.

We thank the Reviewer for noticing the lack of precise information on the follow-up period in the text of the manuscript. This is now defined not only in Figure 1 but also in the manuscript. The following sentence was added into the “2. Materials and Methods – 2.1. Study Design” section:

“The median duration of follow-up was 2,029 days [1283-3059]“.

Reviewer 3 Report

Atrial fibrillation (AF) is a complex pathology that requires an integrated approach in both diagnosis and treatment. The article analyzes the causes of the long-term mortality in patients  with AF and/or chronic coronary syndromes (ССS) to improve the stratification of patients and optimal prevention of thrombosis, clinical symptoms and comorbidities

The paper is well written. The text is clear and easy to read. The topic is original. The study parameters, the design and the long-term mortality data make the article up to date. The independent predictors of the long-term all-cause and CV mortality were revealed that may add some crucial data for the correct prevention treatment.

Authors confirmed that mortality of AF without CCS was at least as high as non-AF patients with CCS. CV deaths were more frequent than non-CV deaths in AF patients with CCS compared to those with either AF or CCS alone.

The conclusions fullfill the evidence, arguments are correct. Authors reflected the main position and suggestion. 

Thank you for the work and could you recalculate the p-value for the age of the alive and dead participants in table 1.

Please, give more precise practical recommendations.

Author Response

Atrial fibrillation (AF) is a complex pathology that requires an integrated approach in both diagnosis and treatment. The article analyzes the causes of the long-term mortality in patients with AF and/or chronic coronary syndromes (ССS) to improve the stratification of patients and optimal prevention of thrombosis, clinical symptoms and comorbidities

The paper is well written. The text is clear and easy to read. The topic is original. The study parameters, the design and the long-term mortality data make the article up to date. The independent predictors of the long-term all-cause and CV mortality were revealed that may add some crucial data for the correct prevention treatment.

Authors confirmed that mortality of AF without CCS was at least as high as non-AF patients with CCS. CV deaths were more frequent than non-CV deaths in AF patients with CCS compared to those with either AF or CCS alone.

The conclusions fullfill the evidence, arguments are correct. Authors reflected the main position and suggestion.

Thank you for the work and could you recalculate the p-value for the age of the alive and dead participants in table 1.

Please, give more precise practical recommendations.

We are grateful for the time and effort invested in reviewing our manuscript and for the many thoughtful comments, kind words, and suggestions from the Reviewer. We have considered these comments carefully and feel that we have fully addressed the reviewer’s concerns. We have replied to each comment below and edited the manuscript accordingly. We believe that this input has improved this manuscript. Complying with all the suggestions, we hope that you will find the manuscript suitable for publication in JCM.

We recalculated age and BMI and the p-value for both of them is correct. Reported values as median and IQR due to lack of normal distribution do not appear to differ strongly, however, when mean and SD are included the differences are as shown below.

In the case of BMI - 28.5 (5.2) vs. 29.2 (4.8)

In the case of age - 64.2 (10) vs. 69.4 (10.1)

Moreover, we thank the Reviewer for the suggestion on adding more practical recommendations. The following sentence is now added into the Conclusions section:

“Therefore, we recommend a careful clinical follow-up of AF patients, with a particular emphasis on stroke prevention and modification of CV risk factors.”

Reviewer 4 Report

In the paper “Predicting mortality in patients with atrial fibrillation and obstructive chronic coronary syndrome: The Bialystok Coronary Project” the authors further analyzed a retrospective cohort study of consecutive patients with confirmed or suspected obstructive CCS. The study had been conducted in the Department of Invasive Cardiology of the Medical University of Bialystok, Poland, and its details and the findings of patients with and without atrial fibrillation (AF) were previously published.

In the present study, the authors reported on the long-term mortality in a median follow-up of 5.6 years and the causes of death according to the diagnosis of AF and/or chronic coronary syndromes (CCS). The crude mortality rate was 16.3%. The authors reported that the patients who died during the follow-up were older, more likely male, had lower body mass index and left ventricular ejection fraction, more often with persistent AF, obstructive CCS, advanced CAD, diabetes, chronic kidney disease (CKD), and chronic obstructive pulmonary disease (COPD). Patients who died were less likely to be diagnosed with hypertension and hyperlipidemia but had higher values of CHA2DS2-Vasc score. The mortality rate was 36.8% in AF (+) and CCS (+), 21.9% in AF (+) and CCS (-), 19.5% in AF (-) and CCS (+), and 9.6% in AF (-) and CCS (-).

Cardiovascular deaths were 25.8% in AF (+) and CCS (+), 12.8% in AF (+) and CCS (-), 10.4% in AF (-) and CCS (+), and 4% in AF (-) and CCS (-). I have to note that in cardiovascular deaths 47% in AF (+) and CCS (+), 47% in AF (+) and CCS (-), 41% in AF (-) and CCS (+), and 35.7% in AF (-) and CCS (-) were reported to be due to coronary artery disease.

The authors concluded that among patients referred for elective coronary angiography, both AF and obstructive CCS are strong and independent predictors of the long-term all-cause and CV mortality. The mortality of AF without CCS was at least as high as non-AF patients with CCS. CV deaths were more frequent than non-CV deaths in AF patients with CCS compared to those with either AF or CCS alone.

Looking at the high proportion of cardiovascular deaths related to coronary artery disease in patients CCS (-) it is likely that the subdivision in CCS (-) and (+) is an oversimplification of the clinical history. I suggest deeply discuss the data that I underlined.

The first paragraph (Out of 26,985 patients …were diagnosed with obstructive CCS.) in Results 3.1 Participant Characteristic is not a result, but the patients’ population in study design.

I have some doubts about the p-value in table 1: the age in alive and dead patients is the same and also BMI seems to differ slightly.

Figures 3 and 4 are confusing: I suggest reporting the forest plot both of all-cause mortality and of cardiovascular mortality (the hazard ratio is not necessary because it is shown in the forest plot).

In my opinion, AF (+) and AF (-) have to be inverted in figure 2 in the inner graph.

Author Response

In the paper “Predicting mortality in patients with atrial fibrillation and obstructive chronic coronary syndrome: The Bialystok Coronary Project” the authors further analyzed a retrospective cohort study of consecutive patients with confirmed or suspected obstructive CCS. The study had been conducted in the Department of Invasive Cardiology of the Medical University of Bialystok, Poland, and its details and the findings of patients with and without atrial fibrillation (AF) were previously published.

In the present study, the authors reported on the long-term mortality in a median follow-up of 5.6 years and the causes of death according to the diagnosis of AF and/or chronic coronary syndromes (CCS). The crude mortality rate was 16.3%. The authors reported that the patients who died during the follow-up were older, more likely male, had lower body mass index and left ventricular ejection fraction, more often with persistent AF, obstructive CCS, advanced CAD, diabetes, chronic kidney disease (CKD), and chronic obstructive pulmonary disease (COPD). Patients who died were less likely to be diagnosed with hypertension and hyperlipidemia but had higher values of CHA2DS2-Vasc score. The mortality rate was 36.8% in AF (+) and CCS (+), 21.9% in AF (+) and CCS (-), 19.5% in AF (-) and CCS (+), and 9.6% in AF (-) and CCS (-).

Cardiovascular deaths were 25.8% in AF (+) and CCS (+), 12.8% in AF (+) and CCS (-), 10.4% in AF (-) and CCS (+), and 4% in AF (-) and CCS (-). I have to note that in cardiovascular deaths 47% in AF (+) and CCS (+), 47% in AF (+) and CCS (-), 41% in AF (-) and CCS (+), and 35.7% in AF (-) and CCS (-) were reported to be due to coronary artery disease.

The authors concluded that among patients referred for elective coronary angiography, both AF and obstructive CCS are strong and independent predictors of the long-term all-cause and CV mortality. The mortality of AF without CCS was at least as high as non-AF patients with CCS. CV deaths were more frequent than non-CV deaths in AF patients with CCS compared to those with either AF or CCS alone.

We are grateful for the time and effort invested in reviewing our manuscript and for the many thoughtful comments, kind words, and suggestions from the Reviewer. We have considered these comments carefully and feel that we have fully addressed the reviewer’s concerns. We have replied to each comment below and edited the manuscript accordingly. We believe that this input has improved this manuscript. Complying with all the suggestions, we hope that you will find the manuscript suitable for publication in JCM.

Looking at the high proportion of cardiovascular deaths related to coronary artery disease in patients CCS (-) it is likely that the subdivision in CCS (-) and (+) is an oversimplification of the clinical history. I suggest deeply discuss the data that I underlined.

As we mentioned in the first limitation of our study, patients were examined only at the time of inclusion in the analysis; there was no reevaluation of the clinical status of patients during their long-term follow-up. It is possible that their group placement status changed during the 10-year follow-up. An additional problem that may be behind this proportion is also the high count of garbage codes in total mortality in Poland. This fact is now also included in the revised study limitations section.

The first paragraph (Out of 26,985 patients …were diagnosed with obstructive CCS.) in Results 3.1 Participant Characteristic is not a result, but the patients’ population in study design.

Thank you for this relevant suggestion. Since this is a duplication of information and its location was indeed incorrect, we have removed the redundant parts of this paragraph.

I have some doubts about the p-value in table 1: the age in alive and dead patients is the same and also BMI seems to differ slightly.

Regarding the second reviewer's concerns (which overlap with this comment), we recalculated age and BMI and the p-value for both of them and p-value is correct, however values are slightly different. Reported values as median and IQR due to lack of normal distribution do not appear to differ strongly, however, when mean and SD are included the differences are as shown below.

In the case of BMI - 28.5 (5.2) vs. 29.2 (4.8)

In the case of age - 64.2 (10) vs. 69.4 (10.1)

The values in tables have been corrected.

Figures 3 and 4 are confusing: I suggest reporting the forest plot both of all-cause mortality and of cardiovascular mortality (the hazard ratio is not necessary because it is shown in the forest plot).

As suggested by the reviewer, we have changed the design of Figures 3 and 4. They are now clearly divided into all-cause mortality and cardiovascular mortality, which increases their readability.

In my opinion, AF (+) and AF (-) have to be inverted in figure 2 in the inner graph.

We apologize for missing this. Thank you for noticing our oversight. You are in fact correct and this has been addressed appropriately.

Round 2

Reviewer 2 Report

I have no further queries